# Rapid SO$_2$ emission reductions significantly increase tropospheric ammonia concentrations over the North China Plain

Mingxu Liu[1], Xin Huang[2], Yu Song[1], Tingting Xu[1], Shuxiao Wang[3], Zhijun Wu[1], Min Hu[1], Lin Zhang[4], Qiang Zhang[5], Yuepeng Pan[6], Xuejun Liu[7], Tong Zhu[1]

[1]State Key Joint Laboratory of Environmental Simulation and Pollution Control, Department of Environmental Science, Peking University, Beijing 100871, China

[2]Joint International Research Laboratory of Atmospheric and Earth System Sciences, School of Atmospheric Sciences, Nanjing University, Nanjing 210023, China

[3]State Key Joint Laboratory of Environmental Simulation and Pollution Control, School of Environment, Tsinghua University, Beijing 100084, China

[4]Laboratory for Climate and Ocean–Atmosphere Studies, Department of Atmospheric and Oceanic Sciences, School of Physics, Peking University, Beijing 100871, China

[5]Ministry of Education Key Laboratory for Earth System Modeling, Center for Earth System Science, Institute for Global Change Studies, Tsinghua University, Beijing 100084, China

[6]State Key Laboratory of Atmospheric Boundary Layer Physics and Atmospheric Chemistry (LAPC), Institute of Atmospheric Physics, Chinese Academy of Sciences 100029, Beijing, China

[7]Beijing Key Laboratory of Farmland Soil Pollution Prevention and Remediation, College of Resources and Environmental Sciences, China Agricultural University, Beijing 100193, China

**Correspondence**: Yu Song (songyu@pku.edu.cn), Min Hu (minhu@pku.edu.cn), and Tong Zhu (tzhu@pku.edu.cn)

**Abstract.** The North China Plain has been identified as a significant hotspot of ammonia (NH$_3$) due to extensive agricultural activities. Satellite observations suggest a significant increase of about 30% in tropospheric gas-phase NH$_3$ concentrations in this area during 2008−2016. However, the estimated NH$_3$ emissions decreased slightly by 7% because of changes in Chinese agricultural practices, i.e., the transition in fertilizer types from ammonium carbonate fertilizer to urea, and in the livestock rearing system from free-range to intensive farming. We note that the emissions of sulfur dioxide (SO$_2$) have rapidly declined by about 60% over the recent few years. By integrating measurements from ground and satellite, a long-term anthropogenic NH$_3$ emission inventory, and chemical transport model simulations, we find that this large SO$_2$ emission reduction is responsible for the NH$_3$ increase over the North China Plain. The simulations for the period 2008−2016 demonstrate that the annual average sulfate concentrations decreased by about 50%,which significantly weakens the formation of ammonium sulfate and increases the average proportions of gas phase NH$_3$ within the total NH$_3$ column concentrations from 26% (2008) to 37% (2016). By fixing SO$_2$ emissions of 2008 in those multi-year simulations, the increasing trend of the tropospheric NH$_3$ concentrations is not observed. Both the decreases in sulfate and increases in NH$_3$ concentrations show highest values in summer, possibly

because the formation of sulfate aerosols is more sensitive to $SO_2$ emission reductions in summer than in other seasons. Besides, the changes in $NO_x$ emissions and meteorological conditions both decreased the $NH_3$ column concentrations by about 3% in the studying period. Our simulations suggest that the moderate reduction in $NO_x$ emissions (16%) favors the formation of particulate nitrate by elevating ozone concentrations in the lower troposphere.

## 1 Introduction

Ammonia ($NH_3$) is considered the most important alkaline gas in the atmosphere. On both a global and regional scale, $NH_3$ is mostly emitted from agricultural activities, mainly including fertilization and livestock industry (Bouwman et al., 1997). Gas-phase $NH_3$ can react with ambient sulfuric and nitric acids to form ammonium sulfate/bisulfate and ammonium nitrate aerosols (SNA), which constitute a significant fraction of atmospheric fine particles ($PM_{2.5}$) associated with potential human health impacts (Pope et al., 2009; Seinfeld and Pandis, 2006). Ammonia and ammonium ($NH_4^+$) is ultimately deposited back to the earth surface, contributing to acid deposition and eutrophication (Asman, 1998; Behera et al., 2013; Pozzer et al., 2017).

As a major agricultural country, China is one of the world's largest emitters of $NH_3$, the amount of which (~10 Tg yr$^{-1}$) exceeds the sum of those in Europe (~4.0 Tg yr$^{-1}$) and North America (~4.0 Tg yr$^{-1}$) (Huang et al., 2012; Bouwman et al., 1997; Paulot et al., 2014). Fertilizer application and livestock manure management contribute to nearly 90% of China's $NH_3$ emissions (Huang et al., 2012; Zhang et al., 2018). Until now, $NH_3$ emissions have not been regulated by the Chinese government, although they serve as an important contributor to haze pollution in China.

The North China Plain (the spatial definition of this area is illustrated in Fig. S1) is a hotspot of $NH_3$ loadings, as revealed by satellite detection and ground measurements (Clarisse et al., 2009; Pan et al., 2018). Interestingly, satellite observations over the past decade have shown an increase in tropospheric columns of gaseous $NH_3$ in this area (Warner et al., 2017). But no sensitivity studies have been performed to explain it, especially from a modelling perspective. A long-term bottom-up inventory indicates that $NH_3$ emissions in China have displayed a slightly decreasing tendency (Kang et al., 2016). During 2006−2016, ammonium bicarbonate for crop fertilization was replaced by urea fertilizer (its fraction of application increasing from 60 to 90% of all mineral nitrogen fertilizers). In the meantime, the traditional free-range livestock system was gradually replaced by intensive animal rearing system (i.e., raising livestock in confinement at a high stocking density) in the livestock industry (increasing

from 21% in 2006 to 48% in 2016; shown in Table S1). These changes in agricultural practices have lowered the volatilization rates of $NH_3$ (Kang et al., 2016).

Several studies have proposed that reduction in $SO_2$ emissions or $NO_x$ emissions is an important factor in determining the increase in atmospheric $NH_3$ concentrations on the global and region scales (Warner et al., 2017; Yu et al., 2018; Saylor et al., 2015). Through the widespread use of the flue gas desulfurization in power plants since 2006 in China, $SO_2$ emissions have gradually decreased (Lu et al., 2011; Li et al., 2010). Li et al. (2017) found it was reduced by 70% from the peak year (around 2006) to 2016 based on satellite observations and bottom up methods. Specifically, the initiation of the "Action Plan for Air Pollution Prevention and Control" (referred to as the national "Ten Measures for Air") since 2013 resulted in a rapid reduction of about 50% over recent few years, from ~30 Tg in 2012 to ~14 Tg in 2016 according to the Multi-resolution Emission Inventory for China (MEIC) (Zheng et al., 2018). To our knowledge, such a strong decrease in $SO_2$ emissions is only found in China. In contrast, emissions of nitrogen oxides ($NO_x$) in MEIC peaked around 2012 with only a moderate decrease of ~20% from 2012 to 2016 (Liu et al., 2016).

Here, we hypothesize that the rapid $SO_2$ emission reduction is the main cause of the increase in tropospheric $NH_3$ concentrations over the North China Plain. To verify this, we first used observation datasets from the ground and space to infer the relationship between the trends in $NH_3$ and $SO_2$ concentrations. A comprehensive long-term $NH_3$ emission inventory, developed by our recent studies based on bottom-up methods, was also used to demonstrate the inter-annual variations of $NH_3$ emissions in this region. Then, we performed multi-year simulations with a chemical transport model to examine the impact of changes in $SO_2$ emissions on tropospheric $NH_3$ concentrations in terms of the magnitude and seasonal variation. Besides, other potential mechanism ($NO_x$ emission and meteorology) were discussed.

## 2 Methods

### 2.1 Observations datasets

Observations from space and ground stations were used in this study. Tropospheric vertical column densities (VCDs) of $NH_3$ were derived from the measurements of Infrared Atmospheric Sounding Interferometer (IASI) onboard MetOp-A (Van Damme et al., 2015; Clarisse et al., 2009; Van Damme et al., 2017). We determined the annual averages of $NH_3$ column concentrations over the North China Plain on a $0.25^{o} \times 0.25^{o}$ grid during 2008−2016, based on the relative error weighting mean method

(Van Damme et al., 2014). The monthly $NH_3$ concentrations were measured using passive $NH_3$ diffusive samplers (Analysts, CNR-Institute of Atmospheric Pollution, Roma, Italy) from September 2015 to August 2016 at 11 sites over Northern China (Pan et al., 2018). The $SO_2$ VCDs were provided by the ozone monitoring instrument (OMI) measurements to test the trend of $SO_2$ concentrations. They were derived from the daily level 3 data set OMSO2e, released by the NASA Goddard Earth Sciences Data and Information Services Center. Besides, daily $PM_{2.5}$ were sampled by quartz-fiber filters at an urban atmosphere environment monitoring station in Peking University (39.99 °N, 116.3 °E) of Beijing, China since 2013. The major water-soluble inorganic compounds (e.g., $NH_4^+$, $NO_3^-$, and $SO_4^{2-}$) were analyzed by ion-chromatography.

## 2.2 WRF-Chem simulations

In this study, the simulations with Weather Research and Forecast Model coupled Chemistry (Grell et al., 2005) version 3.6.1 (WRF-Chem) were conducted for the domain of North China Plain for the years 2008, 2010, 2012, 2014, 2015, and 2016 (referred to as Run_08−16). We ran the model with a horizontal resolution of 30 × 30 km and 24 vertical layers, extending from the surface to 50 hPa. The initial and boundary meteorological condition was derived from 6-h National Centers for Environmental Prediction reanalysis data. The detailed model configuration were described in our previous study (Huang et al., 2014). The anthropogenic emissions from power plant, industrial, residential, and vehicle sectors were taken from the MEIC database. The MEIC data show that the annual $SO_2$ emissions in North China Plain were reduced by about 60%, from 9.9 Tg in 2008 to 4.2 Tg in 2016, while $NO_x$ emissions first increased from 8.0 to 8.8 Tg during 2008−2012, then decreased to 6.7 Tg in 2016.

## 2.3 $NH_3$ emission inventory

A high-resolution $NH_3$ emission inventory (1km×1km, month) was developed based on the bottom-up method. The emission factors were parameterized with regional farming practices, ambient temperature, soil pH and wind speeds etc. The full details can be found in studies by Kang et al. (2016), Huang et al. (2012), and Huo et al. (2015). The inventory has similar spatial features with recent satellite observations (Van Damme et al., 2014), and its amount is close to the emission estimated by the inversion model using ammonium wet deposition data (Paulot et al., 2014). Recent modeling results also showed its good performance by comparing with ammonium observations in China (Huang et al., 2015). The inventory has covered the period from 1980 to 2016 and considered the inter-annual variability in activity levels and agricultural practices. It shows distinct seasonal feature in $NH_3$

emissions over the North China Plain. There are 75% of annual $NH_3$ emissions released in spring and summer months (March-September), during which intensive agricultural fertilization and elevated ambient temperature facilitate the volatilization rates of $NH_3$. Moreover, to integrate this inventory into WRF-Chem simulations, we adopted a diurnal profile with 80% of $NH_3$ emissions in the daytime, following previous studies (Zhu et al., 2015; Asman, 2001; Paulot et al., 2016).

## 3 Results and Discussions

### 3.1 Trends in emissions and concentrations of $NH_3$ vs. $SO_2$

According to the measurements by IASI, the North China Plain showed the highest VCDs of $NH_3$ in China, which mostly ranged from 15 to $30 \times 10^{15}$ molecules $cm^{-2}$ during 2008−2014, and increased to above $30 \times 10^{15}$ molecules $cm^{-2}$ in 2015 and 2016 (Fig. S1). We found the annual $NH_3$ column concentrations increased significantly (*p value* < 0.05) over the North China Plain between 2008 and 2016 (Fig. 1a). The average tropospheric $NH_3$ columns first fluctuated between 2008 and 2013, and then rapidly increased from $21 \times 10^{15}$ molecules $cm^{-2}$ in 2013 to $27 \times 10^{15}$ molecules $cm^{-2}$ in 2016. It showed an overall increase of 30%, or an average annual increase of $0.9 \times 10^{15}$ molecules $cm^{-2}$ $yr^{-1}$. Seasonally, the increase in $NH_3$ columns was more pronounced in summertime (June−August, JJA), with an annual increase rate of $1.8 \times 10^{15}$ molecules $cm^{-2}$ $yr^{-1}$ between 2008 and 2016, which was much higher than in other seasons ($< 1 \times 10^{15}$ molecules $cm^{-2}$ $yr^{-1}$).

In contrast to the trends in tropospheric $NH_3$ concentrations, the annual $NH_3$ emissions first experienced a decreasing tendency from 2008 to 2011 (3.0 Tg in 2009 to 2.8 Tg in 2011), and then remained constant at around 2.8 Tg $yr^{-1}$ during 2011−2016 over the North China Plain (Fig. 1b). The overall trend of $NH_3$ emissions demonstrated a decrease of about 7%. It is because the changes in mineral fertilizer use and livestock rearing practices have lowered $NH_3$ emission rates. The increasing use of urea fertilizer (from 4.7 and 5.2 Tg $yr^{-1}$) and compound fertilizers (from 1.2 to 1.7 Tg $yr^{-1}$) but decreased ammonium bicarbonate (from 1.5 to 0.4 Tg $yr^{-1}$) led to a 20% reduction in $NH_3$ emissions from fertilizer application during 2008−2016 (Table S1). On the other hand, the number of some major livestock increased (Beef −20%, Dairy +39%, Goat −23%, sheep +55%, Pig +18%, and Poultry +19%; see Table S1 for details), while the proportion of intensive animal rearing systems rises to nearly half of the livestock industry in 2016, compared to only 28% in 2008 (Table S1). The intensive systems are characterized with more effective livestock manure management in favor of lower volatilization rates of $NH_3$ (Kang et al., 2016). The transition from the free-range to the intensive in livestock animal rearing

offset the effect of increased animals on the $NH_3$ emissions, thereby resulting in the annual livestock emissions in the North China Plain almost constant (around 1.2 Tg $yr^{-1}$). Overall, the decreasing $NH_3$ emissions cannot track the upward trend of tropospheric $NH_3$ concentrations.

During 2008−2016, $SO_2$ column concentrations were subject to a dramatic decline ($p < 0.01$) due to a 60% decrease in $SO_2$ emissions. The annual mean $SO_2$ VCDs was reduced from $14 \times 10^{15}$ molecules $cm^{-2}$ (2008) to $4 \times 10^{15}$ molecules $cm^{-2}$ (2016), showing a percent reduction of nearly 70%. Especially during 2012−2016, the decreases in $SO_2$ emissions and VCDs accelerated owing to the implementation of the "Action Plan for Air Pollution Prevention and Control" by the Chinese government (Zheng et al., 2018). The ground measurements in a typical urban station in the North China Plain indicated that the annual average $SO_4^{2-}$ concentration in $PM_{2.5}$ decreased by 35% (2013−2016) along with rapid $SO_2$ reductions, which was accompanied by a 33% decrease of particulate $NH_4^+$ (Fig. 1b). Seasonally, the decrease in ground-level $SO_4^{2-}$ reached 60% during summertime (JJA), which was much higher than in other seasons.

## 3.2 Simulations of increasing trend in $NH_3$ columns

We performed numerical simulations with WRF-Chem to interpret the cause of the $NH_3$ increase. We first evaluated model results against measurements of surface $NH_3$ concentrations available in North China Plain as well as the satellite-retrieved $NH_3$ columns. The simulated monthly averaged surface $NH_3$ concentrations at 11 stations (mean + standard deviation: $13.5 \pm 6.8$ μg $m^{-3}$) generally agreed with corresponding observations ($13.4 \pm 9.7$ μg $m^{-3}$) with a correlation coefficient of 0.57. More than 70% of the comparisons differed within a factor of two (Fig. 2). Both simulations and observations show high $NH_3$ concentrations of about 30 μg $m^{-3}$ in warm seasons (March-October) due to enhanced $NH_3$ volatilization and frequent fertilization activities, and lower values (mostly < 15 μg $m^{-3}$) in other months (Fig. 3). Spatially, the hotspot of $NH_3$ was mainly concentrated in Hebei, Shandong and Henan provinces, which have the most intensive agricultural productions in China and thus emit considerable gas-phase $NH_3$ into the atmosphere. We note that the simulated monthly $NH_3$ concentrations were underestimated by 25−70% in several stations in wintertime (January, February, and December). Recently, $NH_3$ emissions from the residential coal and biomass combustion for heating are considered to be a potentially important source of $NH_3$ in suburban and rural areas during wintertime (Li et al., 2016), but it has not been fully included in our bottom-up inventory, which was partially responsible for such deviation between the model and observations.

We calculated the $NH_3$ VCDs from the simulations by integrating $NH_3$ molecular concentrations from

the surface level to top troposphere. The results agreed well with the observed $NH_3$ columns of 2016 on the magnitude and spatial-temporal patterns (Fig. S2). Both IASI measurements and the WRF-Chem simulation showed high annual mean $NH_3$ column concentrations in Hebei, Shandong and Henan provinces, reaching above $30 \times 10^{15}$ molecules $cm^{-2}$. Moreover, we evaluated the modelled SNA concentrations using the filter-based $PM_{2.5}$ samples at an urban atmospheric monitoring station in North China Plain during 2014−2016 (Fig. S3). The model generally reproduced the observed SNA concentrations, with small annual mean bias for sulfate (−2%) and ammonium (−13%) and a relatively large bias for nitrate (−24%). Overall, the model performed well in modelling the concentrations in tropospheric $NH_3$ as well as secondary inorganic aerosols, which provides high confidence for the following interpretation of the $NH_3$ increases.

The model successfully reproduced the observed increasing trend in $NH_3$ columns over the North China Plain during 2008−2016 (Fig. 4). The modelled $NH_3$ columns were systemically lower than the measurements because the relative error weighting mean method always biased a high result due to the smaller relative error in a larger column (Van Damme et al., 2014; Whitburn et al., 2016). An overall increase of 39% in $NH_3$ columns with an average annual increase of $0.8 \times 10^{15}$ molecules $cm^{-2}$ $yr^{-1}$ was found in the simulations between 2008 and 2016, and meanwhile the $SO_2$ columns averaged over the North China Plain decreased by approximately 50% in this period. These results were close to the measurements.

To verify our hypothesis, we replaced $SO_2$ emissions during 2010−2016 by those in 2008, and repeated the simulations (referred to as Run_10_S08 to Run_16_S08). It was noticeable that under these conditions, the increasing trend of $NH_3$ column concentrations was not observed, and even a decrease of 13% took place (Fig. 4). The largest differences were found in 2015 and 2016, when the annual $NH_3$ columns in these sensitivity simulations were about 40% (8−10 $\times 10^{15}$ molecules $cm^{-2}$) lower than those in the baseline cases, corresponding to the 60% reduction in $SO_2$ emissions between 2008 and 2016. By comparing the results among Run_08, Run_16, and Run_16_S08, we found that the reduction in $SO_2$ emissions increased the $NH_3$ column concentrations by 52% during 2008−2016, which was even higher than the overall increase (39%) in the baseline cases. Therefore, we deduce that the rapid $SO_2$ emission reductions are responsible for the increased $NH_3$ levels during 2008−2016, while other mechanisms may be negative contributors. More details on these effects are shown in the following.

**3.3 Influence of $SO_2$ emission reductions on tropospheric $NH_3$ concentrations**

As we indicated above, $SO_4^{2-}$ was observed to be decreasing over recent years in response to the

reductions of $SO_2$ emissions. This was also reproduced by our simulations, which showed that the annual average sulfate concentrations decreased by almost 50% in the lower troposphere. This decreasing trend was especially pronounced after 2013 owing to the much effective $SO_2$ emission reductions. Given that the vapor pressure of $H_2SO_4(g)$ is practically zero over atmospheric particles, atmospheric $SO_4^{2-}$ is predominately in the particle phase and can combine with $NH_3$ available in air, forming sulfate salts (mostly ammonium sulfate/bisulfate) (Seinfeld and Pandis, 2006). Since North China Plain is typically under rich $NH_3$ regimes, $SO_4^{2-}$ is mainly in the form of ammonium sulfate (Meng et al., 2011; Huang et al., 2017); and the aforementioned $SO_4^{2-}$ reductions would therefore increase atmospheric $NH_3$ concentrations by driving the phase state of $NH_3$ from particulate to gaseous.

By assuming that a 1 mol decrease in simulated $SO_4^{2-}$ would lead to a 2 mol increase in ambient gaseous $NH_3$ in this region, the average annual increase in the tropospheric $NH_3$ columns due to the reductions of $SO_4^{2-}$ was estimated to be approximately $1.5 \times 10^{15}$ molecules $cm^{-2}$ $yr^{-1}$ over North China Plain during 2008−2016. This is comparable with or higher than the simulated results from Run_08 to Run_16, as well as the IASI observations ($0.9 \times 10^{15}$ molecules $cm^{-2}$ $yr^{-1}$). By neglecting the deposition processes, we found that the rapid $SO_2$ emission reduction of 50% from 2012 to 2016 resulted in a 55% increase in the $NH_3$ columns, compared to that of 30% recorded by IASI observations. Overall, the estimation results confirmed that the increasing trend of $NH_3$ was mainly determined by the $SO_2$ emission reductions.

We compared the spatial patterns of decreased $SO_4^{2-}$ and increased $NH_3$ between 2008 and 2016 (Run_08 vs. Run_16). Large reductions of $6−10 \times 10^{15}$ molecules $cm^{-2}$ in annual averages of sulfate columns were concentrated in Hebei, Shandong and Henan provinces, the area subject to high $SO_2$ loadings and stringent emission controls (Fig. 5a). Meanwhile, the simulated increases in $NH_3$ columns reached more than $8 \times 10^{15}$ molecules $cm^{-2}$ in most parts of the North China Plain (Fig. 5b), and were comparable with those observed by the IASI ($8−16 \times 10^{15}$ molecules $cm^{-2}$). In addition, we found that $NH_4^+$ concentrations have decreased with a similar magnitude of the increases in gas-phase $NH_3$ levels between Run_08 and Run_16. The proportion of $NH_3$ in the total ($NH_3 + NH_4^+$) increased on average from 26% in 2008 to 37% in 2016 over North China Plain. Figure 5c, d illustrated that without the large $SO_2$ emission reductions between 2008 and 2016 (i.e., replacing $SO_2$ emissions in 2016 by those in 2008, Run_08 vs. Run_16_S08), the sulfate columns partly increased. Correspondingly, the $NH_3$ columns remained constant or decreased by about $5 \times 10^{15}$ molecules $cm^{-2}$ (−13% relative to the 2008 level) in parts of the North China Plain. Thus, the increase in the tropospheric $NH_3$ columns was the result of a transition in $NH_3$ phase partitioning, which was strongly associated with the decreased

formation of ammonium sulfate due to $SO_2$ emission reductions.

The seasonal variations in $SO_4^{2-}$ decreases and $NH_3$ increases were consistent (Fig. 6). We can see that the reduction of sulfate column concentrations between the Run_08 and Run_16 reached $1.3 \times 10^{15}$ molecules $cm^{-2}$ in summer (JJA), which was about three times larger than in other seasons. The corresponding percent reductions ranged from 15% in DJF to 36% in JJA. As aforementioned, the observations of $PM_{2.5}$ in Beijing also showed the highest decrease of sulfate in summer. Considering that the $SO_2$ emission reductions were uniform throughout the year, this seasonal pattern was likely attributed to the conversion efficiency of $SO_2$ to $H_2SO_4$. Our simulations showed that a 1 mol decrease in $SO_2$ corresponded to an approximately 0.7 mol decrease in particulate sulfate in summer over North China Plain, but the values dropped to below 0.4 in other seasons. It is known that the photochemical oxidation of $SO_2$ by OH radical is most active in summertime due to high atmospheric oxidizing capacity, and it dominates the formation of $SO_4^{2-}$, which makes the response of $SO_4^{2-}$ concentrations to $SO_2$ emission reductions more sensitive (Paulot et al., 2017; Huang et al., 2014). The comparison of modelled $NH_3$ columns also showed a markedly higher increase in summer months than during other seasons, driven by the variations in $SO_4^{2-}$. Furthermore, by comparing the model results between the Run_16 and Run_16_S08 cases, we found that without considering the $SO_2$ emission reductions, the seasonal increases in $NH_3$ columns and decreases in $SO_4^{2-}$ concentrations were not observed.

Since the chemical formation of particulate ammonium nitrate also affects the gas-particle partitioning of $NH_3$, the role of $NO_x$ emissions should be discussed. We noted that unlike the trend of particulate sulfate in $PM_{2.5}$, the simulated concentrations of particulate nitrate in $PM_{2.5}$ increased on average by 28% over the North China Plain between 2008 and 2016, despite a 16% reduction in $NO_x$ emissions (Fig. S4). This trend can be partially explained by the increased $NH_3$ in the atmosphere that would facilitate the formation of ammonium nitrate. To quantitatively understand the effect of $NO_x$ emission on the trend of $NH_3$, we performed a sensitivity experiment by repeating the simulation of 2016 with the $NO_x$ emissions in 2008 (Run_16_08N). By comparing the results among Run_16, Run_16_08N, and Run_08, we found that the reduction in $NO_x$ emissions (16% from 2008 to 2016)) decreased the gaseous $NH_3$ concentrations by about 3% (Fig. S5). Specifically, because the reduced $NO_x$ in this period promoted the transition of ozone ($O_3$) photochemistry from VOC-limited to transitional regime with high $O_3$ production efficiency (Jin and Holloway, 2015), the simulated annual mean $O_3$ concentrations were elevated by 3.7 ppb over the North China Plain between the Run_16_08N and Run_16 cases. The resultant enhancement in atmospheric oxidizing capacity would favor the conversion of $NO_2$ to $NO_3^-$ and therefore derive more $NH_3$ partitioning from gaseous to particulate

phases via aerosol thermodynamic equilibrium. Moreover, the measurements at an urban station of Beijing indicated a fluctuating trend of the annual mean $NO_3^-$ concentrations during 2013−2016 (Fig. 1). Overall, the limited reduction in $NO_x$ emissions cannot be responsible for the increased $NH_3$, because the concentrations of particulate nitrate remain high over the North China Plain during recent years.

Besides, meteorological conditions are known to have an influence on $NH_3$ concentrations. Both Warner et al. (2017) and Fu et al. (2017) have found that elevated annual surface temperature partially contributed to the increase in $NH_3$ in East China over the past decade. In this work, we tested the effects of meteorological conditions on $NH_3$ variations by a simulation with meteorological fields in 2016 and anthropogenic emissions in 2012 (Run_12_M16). We selected these two years because $NH_3$ concentrations experienced a rapid increase during the period. This change in meteorological fields for the Run_12_M16 resulted in a decrease of about 3% in annual mean $NH_3$ concentrations relative to the Run_12 (Fig. S6). Therefore, the inter-annual variability in meteorological conditions cannot explain the observed significant increase over the North China Plain.

Interestingly, increasing trends of gas-phase $NH_3$ in the atmosphere have also been observed in the last twenty years in the Midwest of the United States and Western Europe by satellite retrievals and ground measurements (Saylor et al., 2015; Warner et al., 2017; Ferm and Hellsten, 2012). The marked decreases in $SO_2$ and $NO_x$ emissions were largely responsible for these increases, as confirmed by the corresponding trends of particulate sulfate and nitrate concentrations. Warner et al. (2017) infer that $SO_2$ emission reduction in China may be a leading cause of the increased $NH_3$. More recently, Yu et al. (2018) quantified the contributions of the acid gases on the trends of $NH_3$, and found that emissions of $SO_2$ contributed to 2/3 and $NO_x$ to 1/3 of the change in $NH_3$ over the United States from 2001 to 2016. In this work, we demonstrate that the rapid reduction in $SO_2$ emissions was responsible for the increase in $NH_3$ over the North China Plain during 2008−2016, while other potential pathways ($NH_3$ emissions, $NO_x$ emissions, and meteorological conditions) decreased its concentrations by approximately 13% for this period.

## 4 Conclusion

By integrating chemical model simulations and ground and satellite observations, this study investigates an increase (~30%) in tropospheric $NH_3$ column concentrations that has been observed from the space over the North China Plain during 2008−2016. First, the long-term $NH_3$ emission inventory presents a decreasing tendency of −7% in the emission, and therefore it cannot explain the $NH_3$ increase. The meteorological variations and the change in $NO_x$ emissions in the studying period decreased the $NH_3$

column concentrations both by about 3%. Our work strongly indicates that the rapid $SO_2$ emission reductions (60%) from 2008 to 2016 were responsible for the $NH_3$ increase. The multi-year WRF-Chem simulations capture the increasing trend of $NH_3$ and decreasing trend of particulate sulfate well. Simulation results demonstrate that the $SO_2$ emissions reductions decreased the regional mean $SO_4^{2-}$ concentrations by about 50% in the lower troposphere, which reduced the formation of ammonium sulfate particles and consequently increased the average proportions of gas phase $NH_3$ from 26% (2008) to 37% (2016) within the total $NH_3$ column concentrations. The sensitivity simulations by fixing $SO_2$ emissions of 2008 show that without the reductions in $SO_2$ emissions, the increase in $NH_3$ is not observed during 2008−2016, and even a decrease of 13% takes place, which is associated with the effects of other mechanisms ($NH_3$ emissions, $NO_x$ emission, and meteorology). Seasonally, both simulation and observations show the highest decrease in sulfate concentrations, making the increasing trend of $NH_3$ more pronounced in this season. This is likely due to a more sensitive response of sulfate concentrations to $SO_2$ emission reductions in summer than in other seasons.

Given the on-going stringent controls on $SO_2$ emissions in China, a continued increase in $NH_3$ concentrations is anticipated if $NH_3$ emissions are not regulated. The increased tropospheric $NH_3$ levels may have a significant impact on air pollution and nitrogen deposition in China. For instance, the elevated $NH_3$ would facilitate ammonium nitrate formation based on the aerosol thermodynamic equilibrium and negatively impact $PM_{2.5}$ control. That is supported by the fact that $NO_3^-$ concentrations remain high in Northern China and have become increasingly important in contributing to $PM_{2.5}$ pollution (Wen et al., 2018; Li et al., 2018), despite a moderate $NO_x$ emission reduction. The increased proportion of gas-phase $NH_3$ within the total can increase ammonium-nitrogen deposition since gas-phase ammonia deposits more rapidly than particle ammonium. This may alter the spatial pattern of regional nitrogen deposition with higher levels of $NH_3$ deposited near emission sources. These effects are important for human and ecosystem health and need to be investigated in future studies.

*Data availability.* $NH_3$ vertical column density data are freely available through the AERIS database: http://iasi.aeris-data.fr/NH3/. The $SO_2$ vertical column density retrieved from the Ozone Monitoring Instrument is available from Level-3 Aura/OMI Global OMSO2e Data Products released by NASA Goddard Earth Science Data and Information Service Center (https://disc.sci.gsfc.nasa.gov/). Anthropogenic emissions in industry, power plants, transportation, and residential sectors are obtained from Multi-resolution Emission Inventory for China (MEIC, http://www.meicmodel.org/). The PKU-$NH_3$ emission inventory is freely available from the corresponding author Y.S.

(songyu@pku.edu.cn) upon reasonable request.

*Author contributions.* Y.S., M.H., and T.Z. designed the study. Z.W. and M.H. conducted measurements of aerosol chemical compositions. Y.P. conducted measurements of gas-phase ammonia concentrations. Q.Z. developed the MEIC emission database. M.L., X.H. and X.L. contributed to the development of ammonia emission inventory. M.L., X.H., Y.S., T.X., S.W., L.Z and T.Z. analyzed data. M.L. led the writing with input from all co-authors.

*Competing interests*. The authors declare that they have no conflict of interest.

*Acknowledgments.* This study was funded by National Key R&D Program of China (2016YFC0201505), National Natural Science Foundation of China (NSFC) (91644212 and 41675142), and National Research Program for Key Issues in Air Pollution Control (DQGG0208).

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

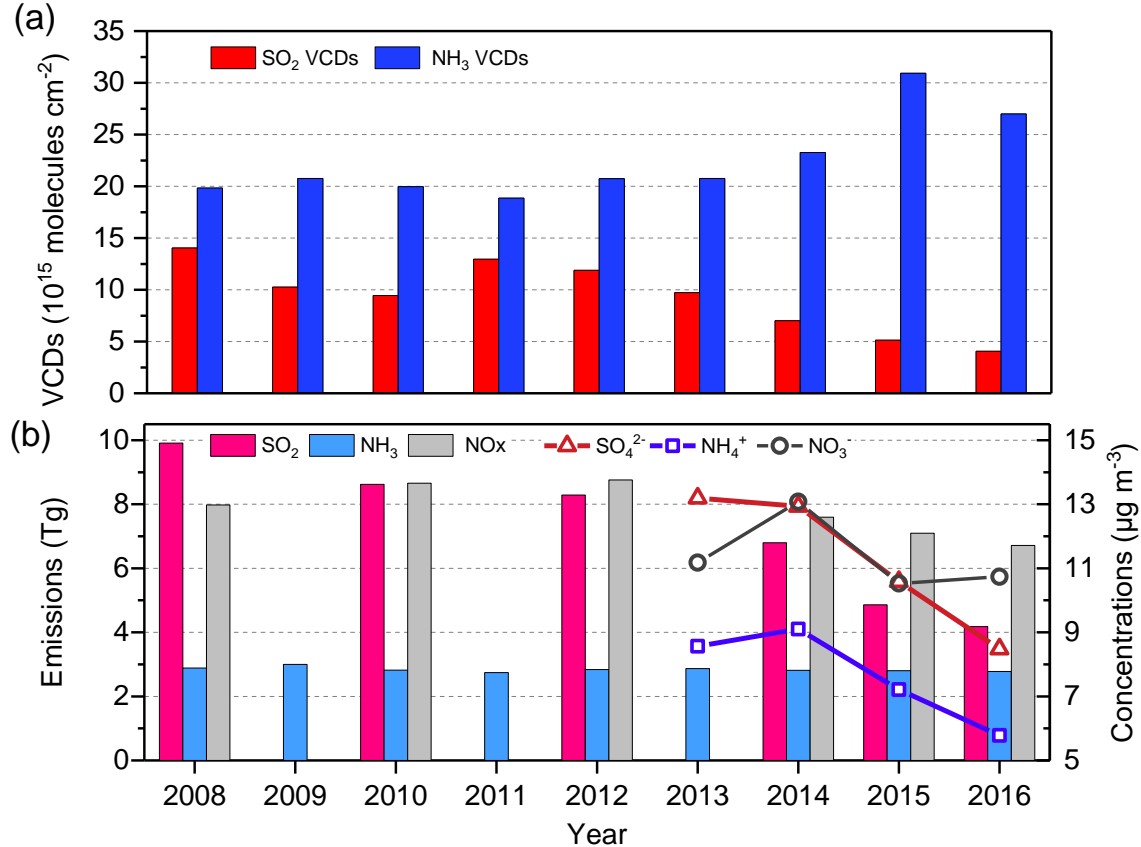

**Figure 1.** (a) Inter-annual trends of $SO_2$ and $NH_3$ VCDs averaged over North China Plain from 2008 to 2016. (b) Inter-annual trends of emissions of $SO_2$ $NH_3$, and $NO_x$ in the North China Plain from 2008 to 2016, and annual mean concentrations of $PM_{2.5}$ sulfate, ammonium, and nitrate derived from measurements at an urban station (Beijing, 39.99 °N, 116.3 °E) in North China Plain from 2013 to 2016.

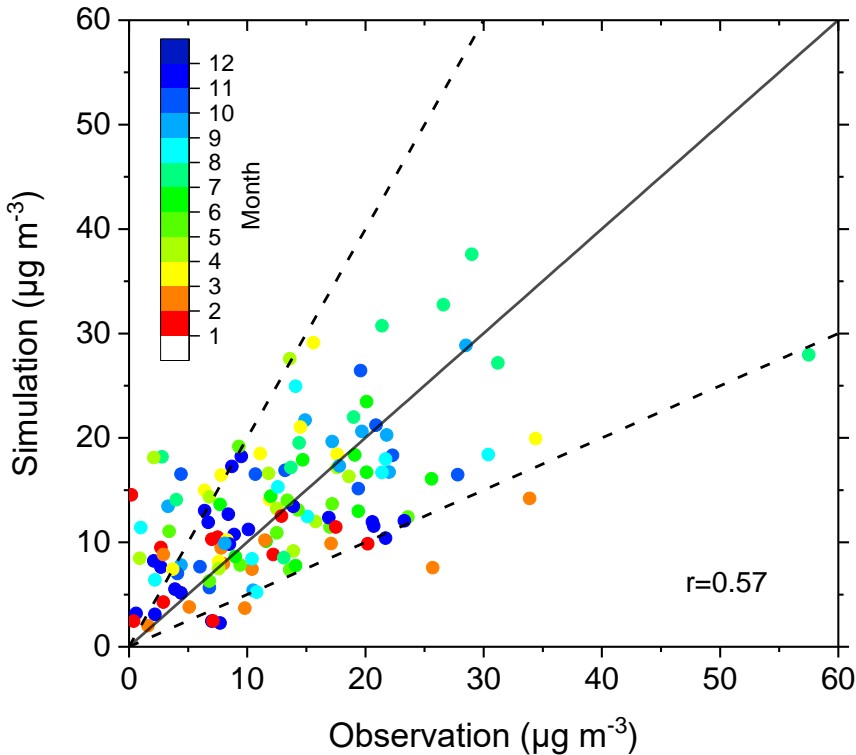

**Figure 2.** Comparison of modelled gaseous NH$_3$ concentrations with corresponding monthly measurements of NH$_3$ from September 2015 to August 2016. The 1:2 and 2:1 dashed lines are shown for reference and the Pearson correlation coefficient (r) is shown inset.

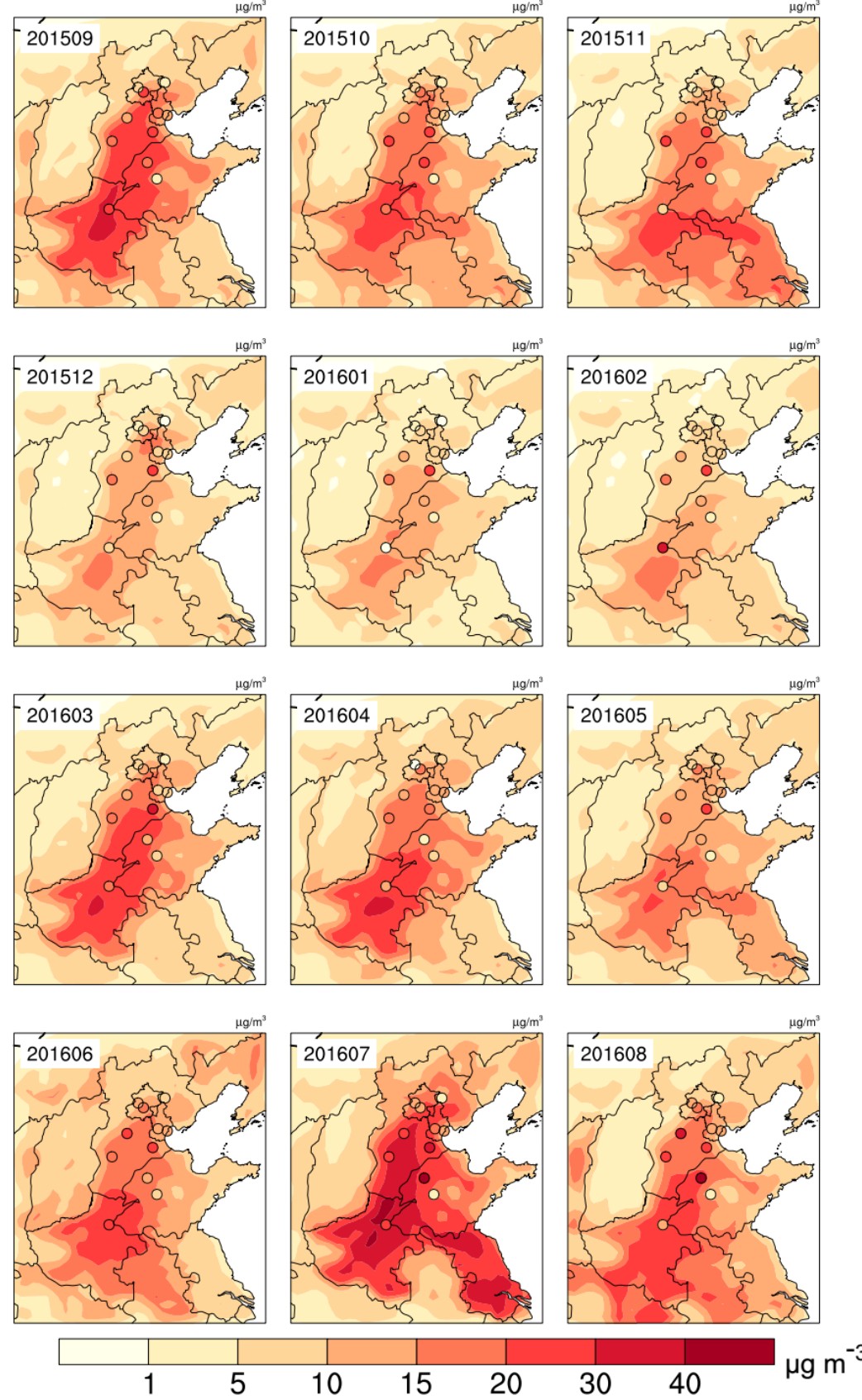

**Figure 3.** Spatial distribution of modelled ground NH$_3$ concentrations ($\mu$g m$^{-3}$) and monthly measurements over North China Plain from September, 2015 (201509) to August, 2016 (201608).

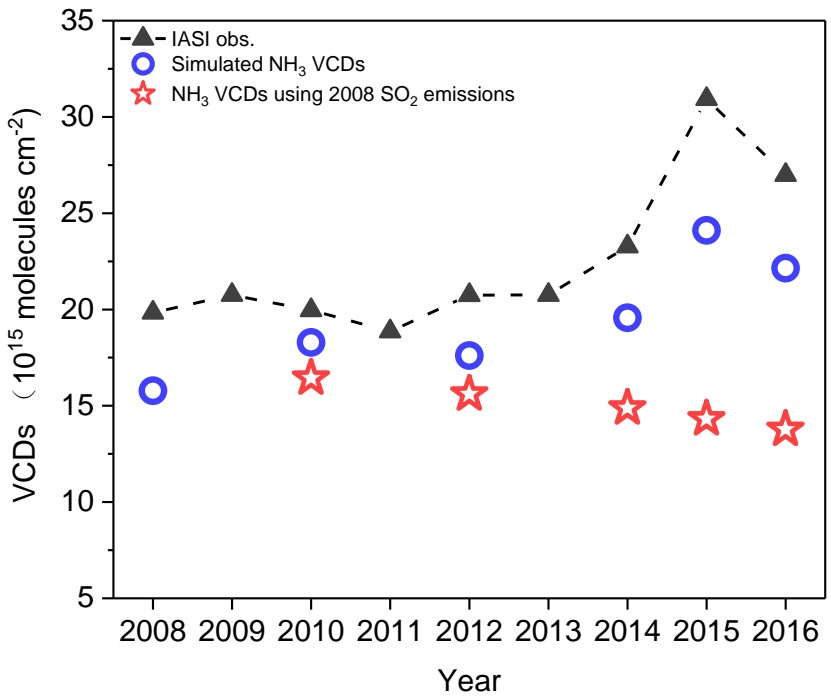

**Figure 4.** Trends in the annual averages of observed and simulated NH$_3$ columns over the North China Plain. The red stars denote the simulated NH$_3$ columns under the 2008 SO$_2$ emissions levels (i.e., Run_10_S08 to Run_16_S08).

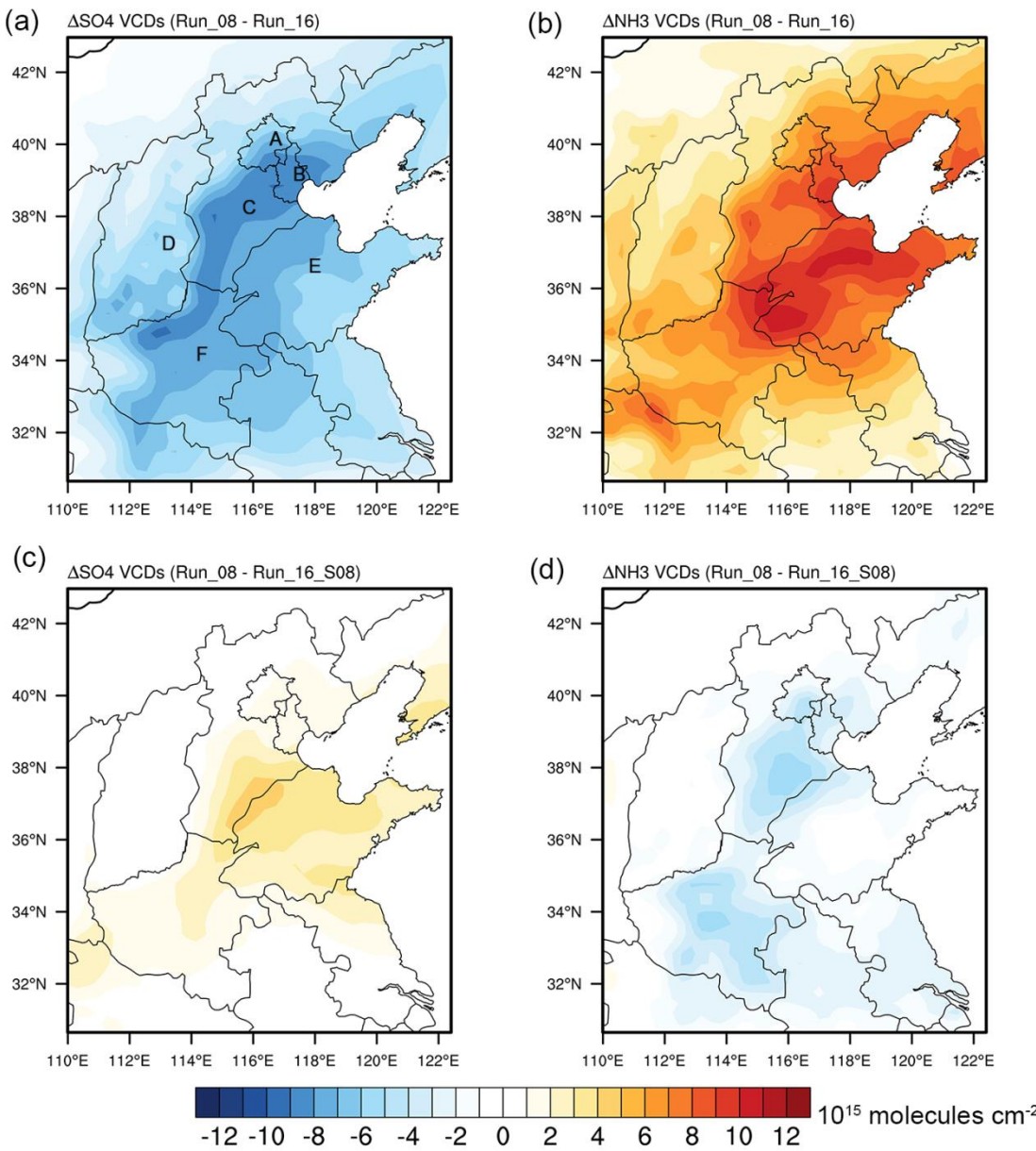

**Figure 5.** The differences between Run_08 and Run_16 (a, b), and between Run_08 and Run_16_S08 (c, d). A-F in Figure. 3a denote Beijing, Tianjin, Hebei, Shanxi, Shandong, and Henan Provinces, respectively.

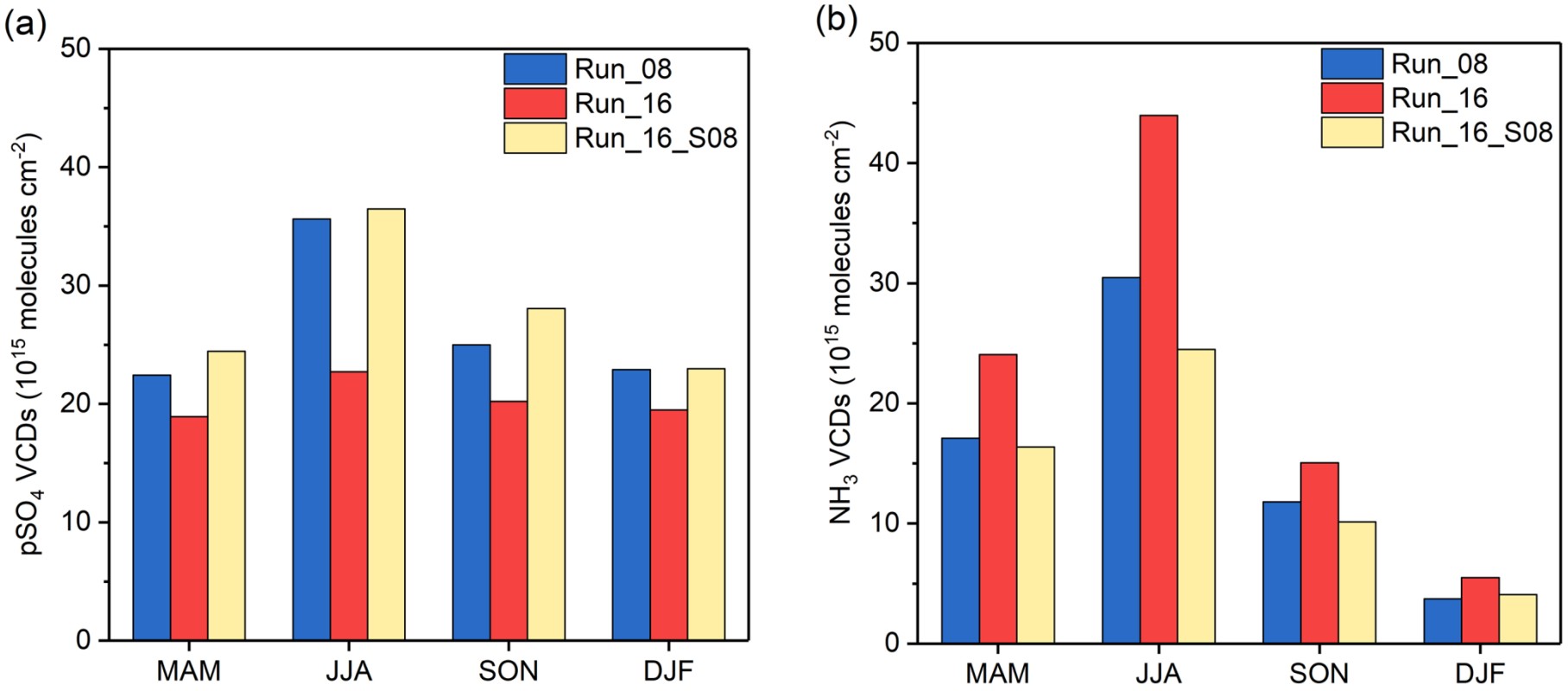

**Figure 6.** Seasonal patterns of simulated $SO_4^{2-}$ (a) and $NH_3$ (b) columns for Run_08, Run_16, and Run_16_S08 (the simulation for 2016 with $SO_2$ emissions in 2008) cases. MAM, JJA, SON and DJF represent spring (March, April and May), summer (June, July and August), autumn (September, October and November) and winter (December, January and February) months.