# Peer review of "Rapid SO2 emission reductions significantly increase tropospheric ammonia concentrations over the North China Plain"

_Atmospheric Chemistry and Physics, 2018_

## Referee Comment (RC1) · Anonymous Referee #1 · 16 Oct 2018

General comments (overall quality)

The paper addresses an important issue relating to atmospheric pollution with ammonia, namely the interaction with SO2 emissions. The material is highly relevant to the subject matter covered by the journal and the results represent a useful contribution to knowledge concerning the interaction between tropospheric ammonia and SO2 emissions. The level of English in the manuscript is fulfilling and the length of the submission seems appropriate. The recommendation is for publication of the paper.

Specific comments (individual scientific questions/issues) The manuscript states that intensive farming results in lower volatilization rates of NH3. This may be true, but I

associate intensive farming with an increased number of livestock, which has the opposite effect, i.e. more livestock increases ammonia emissions. Already in the abstract it would be interesting to get an indication of how the number of livestock has changed over time, as this is an important factor when it comes to emissions of ammonia. It would be interesting to get an indication of the increase (in percentage) on page 5, row 5, that states: The number of some livestock increased slightly.

In the background or discussion, it would be interesting to read more about other similar studies (outside of China), relating to the result in this study, e.g. Aneja et al (2003), Agricultural ammonia emissions and ammonium concentrations associated with aerosols and precipitation in the southeast United States, or Ferm & Hellsten (2012), Trends in atmospheric ammonia and particulate ammonium concentrations in Sweden and its causes.

The authors state that "the increase in ammonia concentrations was highest in summer". However I lack some reasoning regarding seasonal variations in ammonia emissions (e.g. more fertilization of the fields, and higher temperatures in summertime). It would be useful also to mention this in the discussion and its implications on the result.

Technical corrections

Page Row 1 24 Remove "s" in "increases" 2 14 Consider changing "NH3 emission has" to "NH3 emissions have" 3 2 Change "2000" to either "year 2000" or only "2000" 4 13 Change "were" to "was" 5 5 Remove "animals" 5 6 Change "system" to "systems" 5 7 Change " The increased livestock animals raised but more effective...." to "Despite increased livestock numbers, more effective...." 6 17 Change "hotpot" till "hotspot", and change "had" to "have" 6 18 Consider changing "over" to "in" and "into the atmosphere". We noted...." 6 20 Consider adding "s" to emission, i.e. "emissions" 6 21 Consider changing to "it has not been fully included...." 7 16 Change "disappeard" to "stopped" 9 22 "increase in troposheric" 10 3 "....the entire NH3 increase."

Figures Not consistent when it comes to the units, sometimes writing "$\mu$g/m3" and

sometimes "$\mu$g m-3", please consistently use the latter.
* * *

---

## Referee Comment (RC2) · Anonymous Referee #2 · 16 Nov 2018

This manuscript addresses an important topic that is ultimately related to the air quality issues in China. The methodology is sound, as similarly done for the US regions by Yu et al. (2018). I believe it should be published after addressing the following major and minor issues.

My major issue is how authors "claim" their results. Their sensitivity studies of (quote) "the SO2 emission reduction of 50% from 2012 to 2016 could results in a 55% increase in the NH3 columns, compared to that of 30% recorded by IASI observations." ... "the increasing trend of NH3 can be entirely attributable to the SO2 emission reductions." (page 8, line 6-12). I do not believe such a conclusion can be drawn, unless the

authors have performed and show quantitatively that all other mechanisms (NOx, NH3 emissions, temperatures, precipitations, etc.) do not contribute to the NH3 increase (see more below). The estimated increase of 55% being larger than the observations of 30% only indicates uncertainties.

The last paragraph before Conclusion (page 9, line 14-22) is ambiguous and hand-waving. These "other" mechanisms that are very likely to have also caused the gaseous NH3 to increase, but were dismissed without sufficient quantitative data or figures to back it up. (quote) ". . .particulate nitrate. . . concentrations appear to increase in the North China Plain between 2008 and 2016 despite a 23% reduction in NOx emission (Fig. S4). The in situ measurements in Beijing indicated that the NO3− concentrations fluctuated during 2013−2016. It implied that the NOx emission reduction could not be responsible for the increase in NH3." Should not "imply" a mechanism that "could not be" responsible. . . The same process for the SO2 should be repeated for the NOx, if any conclusions were to be drawn about how NOx reduction affects the gaseous NH3 concentration change. The in situ measurements in Beijing was used to make an argument, but no evidence was shown in the manuscript, additionally, the where about of the data is not included, which does not follow the ACP data policy.

Similarly, for meteorological effects, quote "We also tested the effects of meteorological conditions on NH3 variations by a simulation with meteorological fields in 2016 and anthropogenic emissions in 2012 (Run_16_E12). Compared to the Run_12 case, we found the change in meteorological fields (2012 vs. 2016) had a negligible influence on NH3 concentrations in most of North China Plain." None of these were shown quantitatively! Can't make statements like these without any evidence. The following statement "Although temperature increase was reported to partly contribute to the positive trend of NH3 (Warner et 20 al., 2017; Fu et al., 2017), our simulations indicated that the overall meteorological factors could not explain the observed significant increase tropospheric NH3 concentrations over North China Plain." This sentence is misleading, as if the quoted studies were trying to explain the observed significant increase in tro-

pospheric NH3 concentrations by meteorological factors. In fact, Warner et al. (2017) emphasized the leading cause of the NH3 increase was the reduction of SO2 in China, I quote "Over China, a combination of expanded agricultural activities, nascent SO2 control measures, and increasing temperatures cause the observed increases in ammonia."

My minor issues are mainly related to language and choice of words. I believe this manuscript needs to go through English editor at ACP. Also, many word choices are not appropriate for concise scientific publications, and somewhat wishy-washy, e.g., "appear to", "could not be", "may be a potential", "could be responsible", "would bias", ". . . concentrations disappeared", ". . . is practically zero. . .", "could result", "were almost consistent", "could make", "implied", "for almost the entire. . .", "not well-regulated", "can increase. . .", "may alter". . .

Page 2 line 11: "As a major agricultural country, China is the world's largest emitter of NH3. . ." what about India?

Page 2 line 15: ". . .may be potentially important contributor to haze. . ." It's a known fact!

Page 2 line 17-19: "Interestingly, satellite observations over the past decade have shown an increase in tropospheric columns of gaseous NH3in this area (Warner et al., 2017). But no quantitative studies have been performed to explain it." Warner et al. (2017) was a quantitative study using observations. Should be "But no sensitivity studies. . ."

Page 2 line 19-20: "Along-term bottom-up inventory indicated that NH3 emissions in China have displayed a slightly decreasing tendency." Needs references!

Page 3 line 10: "Here, we hypothesize that the rapid SO2 emission reduction is the reason for the increase in tropospheric NH3. . ." Several studies have published the fact that the SO2 emission reduction is the reason. . ., not a hypothesis anymore. Should

reference others' publications here, for global studies or in other regions, than in the North China.

Page 4 line 9: Please pay attention to the order when acronyms are introduced and used throughout the paper.

Page 4 line 15: MEIC should be defined on Page 3 line 7.

Page 4 line 15: "were cut" use reduced.

Page 4 line 19: remove "by our research group"

Page 4 line 21: "in our previous studies..." should be "studies by..."

Page 5 line 5-7: "Meanwhile..." needs references.

Page 5 line 11: use IASI.

Page 6 line 22: "which could be responsible", add partially responsible...

Page 6 line 23-24: bad sentence, rewrite.

Page 7 line 3: "Moreover, we also...", remove also.

Page 7 line 18-19: "These tests support..." Too absolute! No other mechanisms?

Fig. 2: use whole words for Sim., Obs., Sep., and Aug.

Yu et al. (2018) "Long-Term Trend of Gaseous Ammonia Over the United States: Modeling and Comparison With Observations" DOI: 10.1029/2018JD028412
* * *

---

## Author Comment (AC1) · 29 Nov 2018

**Response to Referee #1**

*General comments (overall quality):*

*The paper addresses an important issue relating to atmospheric pollution with ammonia, namely the interaction with $SO_2$ emissions. The material is highly relevant to the subject matter covered by the journal and the results represent a useful contribution to knowledge concerning the interaction between tropospheric ammonia and $SO_2$ emissions. The level of English in the manuscript is fulfilling and the length of the submission seems appropriate. The recommendation is for publication of the paper.*

**Response:** We would like to thank the referee for the encouragements and providing the insightful suggestions, which indeed help us to improve the manuscript.

*Specific comments (individual scientific questions/issues):*

*The manuscript states that intensive farming results in lower volatilization rates of $NH_3$. This may be true, but I associate intensive farming with an increased number of livestock, which has the opposite effect, i.e. more livestock increases ammonia emissions. Already in the abstract it would be interesting to get an indication of how the number of livestock has changed over time, as this is an important factor when it comes to emissions of ammonia. It would be interesting to get an indication of the increase (in percentage) on page 5, row 5, that states: The number of some livestock increased slightly.*

**Response:** Accepted. There are an increased number of livestock animals during 2008−2016. We have shown the detailed information in Table S1. As suggested by the referee, we added the quantitative data in the revised manuscript.

The intensive systems are characterized by lower $NH_3$ emission factors than the free-range (traditional animal rearing system in rural area) (Huang et al., 2012; Kang et al., 2016; EEA). The increased number of animals will produce more manure in farms, but applying intensive farming in livestock industry lowers the volatilization rates of $NH_3$ per animal. In this work, we find that the transition from the free-range to the intensive farming in the Chinese livestock industry offset the effect of increased animals on the $NH_3$ emissions. The resulting livestock $NH_3$ emissions in northern China do not show a significant trend during this period.

**Revision:** (Page 5, Line 23-30) "On the other hand, the number of some major livestock increased (Beef −20%, Dairy +39%, Goat −23%, sheep +55%, Pig +18%, and Poultry +19%; see Table S1 for details), while the proportion of intensive animal rearing systems rises to nearly half of the

livestock industry in 2016, compared to only 28% in 2008 (Table S1). The intensive systems are characterized with more effective livestock manure management in favor of lower volatilization rates of $NH_3$ (Kang et al., 2016). The transition from the free-range to the intensive in livestock animal rearing offset the effect of increased animals on the $NH_3$ emissions, thereby resulting in the annual livestock emissions in the North China Plain almost constant (around 1.2 Tg $yr^{-1}$). Overall, the decreasing $NH_3$ emissions cannot track the upward trend of tropospheric $NH_3$ concentrations."

*In the background or discussion, it would be interesting to read more about other similar studies (outside of China), relating to the result in this study, e.g. Aneja et al (2003), Agricultural ammonia emissions and ammonium concentrations associated with aerosols and precipitation in the southeast United States, or Ferm & Hellsten (2012), Trends in atmospheric ammonia and particulate ammonium concentrations in Sweden and its causes.*

**Response:** Accepted. We cited those related papers in the background and discussions parts in the revised manuscript.

**Revision:** (Page 3, Line 1-3) "Several studies have proposed that reduction in $SO_2$ emissions or $NO_x$ emissions is an important factor in determining the increase in atmospheric $NH_3$ concentrations on the global and region scales (Warner et al., 2017; Yu et al., 2018; Saylor et al., 2014)."

(Page 10, Line 12-23) "Interestingly, increasing trends of gas-phase $NH_3$ in the atmosphere have also been observed in the last twenty years in the Midwest of the United States and Western Europe by satellite retrievals and ground measurements (Saylor et al., 2015; Warner et al., 2017; Ferm and Hellsten, 2012). The marked decreases in $SO_2$ and $NO_x$ emissions were largely responsible for these increases, as confirmed by the corresponding trends of particulate sulfate and nitrate concentrations. Warner et al. (2017) infer that $SO_2$ emission reduction in China may be a leading cause of the increased $NH_3$. More recently, Yu et al. (2018) quantified the contributions of the acid gases on the trends of $NH_3$, and found that emissions of $SO_2$ contributed to 2/3 and $NO_x$ to 1/3 of the change in $NH_3$ over the United States from 2001 to 2016. In this work, we demonstrate that the rapid reduction in $SO_2$ emissions was responsible for the increase in $NH_3$ over the North China Plain during 2008−2016, while other potential pathways ($NH_3$ emissions, $NO_x$ emissions, and meteorological conditions) decreased its concentrations by approximately 13% for this period."

*The authors state that "the increase in ammonia concentrations was highest in summer". However I lack some reasoning regarding seasonal variations in ammonia emissions (e.g. more fertilization of the fields, and higher temperatures in*

*summertime). It would be useful also to mention this in the discussion and its implications on the result.*

**Response:** The NH$_3$ emissions in the North China Plain are concentrated in spring and summertime due to frequent fertilization activities and higher temperature, which facilitates the volatilization rates of NH$_3$. We added the description about the seasonal distribution of NH$_3$ emissions in the North China Plain in the Methods of the revised manuscript.

However, the main finding of our study is that the rapid reduction in SO$_2$ emissions in China strongly reduced the concentrations of ammonium sulfate aerosols and transfers NH$_3$ from particle to gas phases. Therefore, the seasonal feature of the increase in gaseous NH$_3$ concentrations was mainly determined by that of sulfate concentrations rather than the NH$_3$ emissions. Both observations and simulation indicate that the concentrations of particulate sulfate demonstrated highest decreases in summer, thereby causing high increase in NH$_3$.

**Revision:** (Page 4, Line 28-30; Page 5, Line 1-3) "It shows distinct seasonal feature in NH$_3$ emissions over the North China Plain. There are 75% of annual NH$_3$ emissions released in spring and summer months (March-September), during which intensive agricultural fertilization and elevated ambient temperature facilitate the volatilization rates of NH$_3$. In this study, to integrate our NH$_3$ inventory into WRF-Chem simulations, we adopted a diurnal profile with 80% of the NH$_3$ emissions in the daytime, following previous studies (Zhu et al., 2015; Paulot e al., 2016; Asman, 2001)."

(Page 9, Line 1-4) "The seasonal variations in SO$_4^{2-}$ decreases and NH$_3$ increases were consistent (Fig. 6). We can see that the reduction of sulfate column concentrations between the Run_08 and Run_16 reached $1.3 \times 10^{15}$ molecules/cm$^2$ in summer (JJA), which was about three times larger than in other seasons. The corresponding percent reductions ranged from 15% in DJF to 36% in JJA. As aforementioned, the long-term observations of PM$_{2.5}$ in Beijing also confirmed the highest decrease of sulfate in summer."

*Technical corrections*

*Page Row 1 24 Remove "s" in "increases"*

**Response:** Accepted. We remove it.

**Revisions:** (Page 1, Line 27) "we demonstrate that this large SO$_2$ emission reduction is responsible for the NH$_3$ increase"

*2 14 Consider changing "NH3 emission has" to "NH3 emissions have"*

**Response:** Accepted. We change it.

**Revisions:** (Page 2, Line 17-18) "Until now, $NH_3$ emissions have not been regulated by the Chinese government, although they serve as an important contributor to haze pollution in China."

*3 2 Change "2000" to either "year 2000" or only "2000"*

**Response:** Accepted. We change it.

**Revisions:** (Page 3, Line 3-4) "Through the widespread use of the flue gas desulfurization in power plants since 2006 in China, $SO_2$ emissions have gradually decreased."

*4 13 Change "were" to "was"*

**Response:** Accepted. We change it.

**Revisions:** (Page 4, Line 20) "A high-resolution $NH_3$ emission inventory (1km$\times$1km, month) was developed based on the bottom-up method."

*5 5 Remove "animals" 5 6 Change "system" to "systems" 5 7 Change "The increased livestock animals raised but more effective...." to "Despite increased livestock numbers, more effective...."*

**Response:** Accepted. We reword these statements.

**Revisions:** (Page 5, Line 23-30) "On the other hand, the number of some major livestock increased (Beef −20%, Dairy +39%, Goat −23%, sheep +55%, Pig +18%, and Poultry +19%; see Table S1 for details), while the proportion of intensive animal rearing systems rises to nearly half of the livestock industry in 2016, compared to only 28% in 2008 (Table S1). The intensive systems are characterized with more effective livestock manure management in favor of lower volatilization rates of $NH_3$ (Kang et al., 2016). The transition from the free-range to the intensive in livestock animal rearing offset the effect of increased animals on the $NH_3$ emissions, consequently resulting in the annual livestock emissions in the North China Plain being almost constant (around 1.2 Tg per year)."

*6 17 Change "hotpot" till "hotspot", and change "had" to "have" 6 18 Consider changing "over" to "in" and "into the atmosphere". We noted...."*

**Response:** Accepted. We reword the statement.

**Revisions:** (Page 6, Line 21-23) "Spatially, the hotspot of $NH_3$ was mainly concentrated in Hebei, Shandong and Henan provinces, which have the most intensive agricultural productions in China and thus emit considerable gas-phase $NH_3$ into the atmosphere."

*6 20 Consider adding "s" to emission, i.e. "emissions"*

**Response:** Accepted. We reword it.

**Revisions:** (Page 6, Line 25) "Recently, $NH_3$ emissions from the residential coal and biomass combustion for heating are considered to be a potentially important source of $NH_3$ in suburban and rural areas during wintertime"

*6 21 Consider changing to "it has not been fully included...."*

**Response:** Accepted. We change it.

**Revisions:** (Page 6, Line 27) "it has not been fully included in our bottom-up inventory."

*7 16 Change "disappeard"*

**Response:** Accepted. We reword the statement.

**Revisions:** (Page 7, Line 18-19) "It was noticeable that under these conditions, the increasing trend of $NH_3$ column concentrations was not observed."

*Figures Not consistent when it comes to the units, sometimes writing "µg/m3" and sometimes "_g m-3", please consistently use the latter.*

**Response:** Accepted. We use the unit of µg m$^{-3}$ in the whole manuscript.

**Revisions:**

[Figure]

**Figure 1.** (a) Inter-annual trends of $SO_2$ and $NH_3$ VCDs averaged over North China Plain from 2008 to 2016. (b) Inter-annual trends of emissions of $SO_2$ $NH_3$, and $NO_x$ in the North China Plain from 2008 to 2016, and annual mean concentrations of $PM_{2.5}$ sulfate, ammonium, and nitrate derived from measurements at an urban station (Beijing, 39.99 °N,

116.3 °E) in North China Plain from 2013 to 2016.

[Figure]

**Figure 2.** Comparison of modelled gaseous NH₃ concentrations with corresponding monthly measurements of NH₃ from September 2015 to August 2016. The 1:2 and 2:1 dashed lines are shown for reference and the Pearson correlation coefficient (r) is shown inset

*References*

Asman, W. A.: Modelling the atmospheric transport and deposition of ammonia and ammonium: an overview with special reference to Denmark, Atmos. Environ., 35, 1969-1983, 2001.

EEA: EMEP/EEA air pollutant emission inventory guidebook 2013, European Environment Agency, 2013

Ferm, M., and Hellsten, S.: Trends in atmospheric ammonia and particulate ammonium concentrations in Sweden and its causes, Atmos. Environ., 61, 30-39, https://doi.org/10.1016/j.atmosenv.2012.07.010, 2012.

Huang, X., Song, Y., Li, M., Li, J., Huo, Q., Cai, X., Zhu, T., Hu, M., and Zhang, H.: A high-resolution ammonia emission inventory in China, Global Biogeochem. Cy., 26, GB1030, 10.1029/2011GB004161, 2012.

Kang, Y., Liu, M., Song, Y., Huang, X., Yao, H., Cai, X., Zhang, H., Kang, L., Liu, X., Yan, X., He, H., Zhang, Q., Shao, M., and Zhu, T.: High-resolution ammonia emissions inventories in China from 1980 to 2012, Atmos. Chem. Phys., 16, 2043-2058, 10.5194/acp-16-2043-2016, 2016.

Paulot, F., Ginoux, P., Cooke, W. F., Donner, L. J., Fan, S., Lin, M. Y., Mao, J., Naik, V., and Horowitz, L. W.: Sensitivity of nitrate aerosols to ammonia emissions and to nitrate chemistry: implications for present and future nitrate optical depth, Atmos. Chem. Phys., 16, 1459-1477, 10.5194/acp-16-1459-2016, 2016.

Saylor, R., Myles, L., Sibble, D., Caldwell, J., and Xing, J.: Recent trends in

gas-phase ammonia and PM2.5 ammonium in the Southeast United States, J Air Waste Manag Assoc, 65, 347-357, 10.1080/10962247.2014.992554, 2015.

Yu, F., Nair, A. A., and Luo, G.: Long-term trend of gaseous ammonia over the United States: Modeling and comparison with observations, J. Geophys. Res. Atmos., 123, 8315-8325, doi:10.1029/2018JD028412, 2018.

Zhu, L., Henze, D., Bash, J., Jeong, G. R., Cady-Pereira, K., Shephard, M., Luo, M., Paulot, F., and Capps, S.: Global evaluation of ammonia bidirectional exchange and livestock diurnal variation schemes, Atmos. Chem. Phys., 15, 12823-12843, 10.5194/acp-15-12823-2015, 2015.

---

## Author Comment (AC2) · 29 Nov 2018

**Response to Referee #2**

*This manuscript addresses an important topic that is ultimately related to the air quality issues in China. The methodology is sound, as similarly done for the US regions by Yu et al. (2018). I believe it should be published after addressing the following major and minor issues.*

**Response:** We would like to thank the referee for the insightful comments. We accepted all the comments and suggestions, and improved the manuscript thoroughly.

*My major issue is how authors "claim" their results. Their sensitivity studies of (quote) "the SO2 emission reduction of 50% from 2012 to 2016 could results in a 55% increase in the NH3 columns, compared to that of 30% recorded by IASI observations." : : : "the increasing trend of NH3 can be entirely attributable to the SO2 emission reductions." (page 8, line 6-12). I do not believe such a conclusion can be drawn, unless the authors have performed and show quantitatively that all other mechanisms (NOx, NH3 emissions, temperatures, precipitations, etc.) do not contribute to the NH3 increase (see more below). The estimated increase of 55% being larger than the observations of 30% only indicates uncertainties.*

**Response:** Accepted. In addition to the evidences for the effect of $SO_2$ reduction on the $NH_3$ increase, we provided quantitative results of other mechanisms in the revised manuscript, as following.

- $NH_3$ emissions. Our inventory has demonstrated that $NH_3$ emissions in northern China experienced an overall decrease of 7% from 2008 to 2016. This decrease is caused by the changes in fertilizer use and livestock rearing practices in farms. The $NH_3$ emissions would decrease its concentrations in this period.

- $NO_x$ emissions. The anthropogenic $NO_x$ emissions in the North China Plain first increased from 2008 to 2012 by 10%, and then decreased by 23% afterwards. The overall trend of $NO_x$ emissions is a decrease of 17% during 2008−2016. However, our simulations indicated an increase of 28% in the mean particulate nitrate concentrations in the region from 2008−2016. It can be explained by the significantly increased $NH_3$ that facilitates the formation of ammonium nitrate as well as enhanced atmospheric oxidizing capacity. We re-run the simulation of 2016 by replacing the $NO_x$ emissions with those in 2008. The results indicate that the change in $NO_x$ emissions between 2008 and 2016 gives rise to a slight decrease in the $NH_3$ column concentrations of about 3%. So it cannot be responsible for the significant increase of $NH_3$.

♦ Meteorological conditions. We did a sensitive simulation with meteorological fields in 2016 and anthropogenic emissions in 2012 (the period of 2012−2016 showing a rapid increase in $NH_3$). The change in meteorological fields between the Run_2012 and Run_12_M16 led to a decrease in $NH_3$ concentrations of ~3% over the North China Plain.

The above mechanisms totally decreased the $NH_3$ column concentrations by about 13%. So we conclude that the $SO_2$ emission reductions is responsible for the increasing trend of $NH_3$. More details for other mechanisms (especially $NO_x$ emissions and meteorology) are shown in the following responses.

**Revisions:** (Page 5, Line 16-19) "the annual $NH_3$ emissions first experienced a decreasing tendency from 2008 to 2011 (3.0 Tg in 2009 to 2.8 Tg in 2011), and then remained constant at around 2.8 Tg during 2011−2016 over the North China Plain (Fig. 1b). The overall trend of $NH_3$ emissions demonstrated a decrease of about 7%."

(Page 9, Line 21-31) "To quantitatively understand the effect of $NO_x$ emission on the trend of $NH_3$, we performed a sensitive experiment by repeating the simulation of 2016 with the $NO_x$ emissions in 2008 (Run_16_08N). By comparing the results among Run_16, Run_16_08N, and Run_08, we found that the reduction in $NO_x$ emissions (17% from 2008 to 2016)) decreased the gaseous $NH_3$ concentrations by about 3% (Fig. S5). Specifically, because the reduced $NO_x$ in this period led to the transition of ozone ($O_3$) photochemistry from VOC-limited to transitional regime with high $O_3$ production efficiency (Jin and Holloway, 2015), the simulated annual mean $O_3$ concentrations were elevated by 3.7 ppb over the North China Plain between the Run_16_08N and Run_16 cases. The resultant enhancement in atmospheric oxidizing capacity would favor the conversion of $NO_2$ to $NO_3^-$ and therefore derive more $NH_3$ partitioning from gas to particle phases via aerosol thermodynamic equilibrium."

(Page 10, Line5-10) "In this work, we tested the effects of meteorological conditions on $NH_3$ variations by a simulation with meteorological fields in 2016 and anthropogenic emissions in 2012 (Run_12_M16). We selected these two years because $NH_3$ concentrations experienced a rapid increase during the period. This change in meteorological fields for the Run_12_M16 resulted in a decrease of 3% in annual mean $NH_3$ concentrations relative to the Run_12 (Fig. S6)."

(Page 10, Line 20-23) "In this work, we demonstrate that the rapid reduction in $SO_2$ emissions was responsible for the increase in $NH_3$ over the North China Plain during 2008−2016, while other potential pathways ($NH_3$ emissions, $NO_x$ emissions, and meteorological conditions) decreased its concentrations by approximately 13% for this period."

(Page 10, Line 27-30) "First, the long-term $NH_3$ emission inventory presents a decreasing tendency of −7% in the emission, and therefore it cannot explain the $NH_3$ increase. The meteorological variations and the change in $NO_x$ emissions in the studying period decreased the $NH_3$ column concentrations both by about 3%."

*The last paragraph before Conclusion (page 9, line 14-22) is ambiguous and handwaving. These "other" mechanisms that are very likely to have also caused the gaseous NH3 to increase, but were dismissed without sufficient quantitative data or figures to back it up. (quote) ": : :particulate nitrate: : : concentrations appear to increase in the North China Plain between 2008 and 2016 despite a 23% reduction in NOx emission (Fig. S4). The in situ measurements in Beijing indicated that the NO3- concentrations fluctuated during 2013-2016. It implied that the NOx emission reduction could not be responsible for the increase in NH3." Should not "imply" a mechanism that "could not be" responsible: : : The same process for the SO2 should be repeated for the NOx, if any conclusions were to be drawn about how NOx reduction affects the gaseous NH3 concentration change. The in situ measurement in Beijing was used to make an argument, but no evidence was shown in the manuscript, additionally, the where about of the data is not included, which does not follow the ACP data policy.*

**Response:** Accepted. As suggested by the referee, we performed another sensitive simulation for 2016 by using $NO_x$ emissions in 2008. The resulting $NH_3$ column concentrations were 2% higher than those in the baseline simulation for 2016. When compared to the 2008 simulation, the reduction in $NO_x$ emissions during 2008−2016 decreased the $NH_3$ concentrations on average by 3%. We provide quantitative results in the revised manuscript and also show the effect of $NO_x$ emissions in Fig. S6.

The measurements of $PM_{2.5}$ chemical components (including sulfate, nitrate, and ammonium) were conducted in Peking University, Beijing since 2013 (please see Section 2.1). We show the inter-annual trend of $PM_{2.5}$ nitrate concentrations in Fig. 1 in the revised manuscript. The annual mean concentrations of nitrate fluctuated during 2013−2016 without a significant trend.

Based on these evidences from the sensitive simulation and the observations, the change in $NO_x$ emission has a negligible contribution on the $NH_3$ increase during 2008−2016.

**Revisions:** (Page 9, Line 16-32) "Since the chemical formation of particulate ammonium nitrate also affects the gas-particle partitioning of $NH_3$, the role of $NO_x$ emissions should be discussed. We noted that unlike the trend of particulate sulfate in $PM_{2.5}$, the simulated concentrations of particulate nitrate in $PM_{2.5}$ increased on average by 28% over the North China Plain between 2008 and 2016, despite a 17% reduction in $NO_x$ emissions (Fig.

S4). This trend can be partially explained by the increased $NH_3$ in the atmosphere that would facilitate the formation of ammonium nitrate. To quantitatively understand the effect of $NO_x$ emission on the trend of $NH_3$, we performed a sensitive experiment by repeating the simulation of 2016 with the $NO_x$ emissions in 2008 (Run_16_08N). By comparing the results among Run_16, Run_16_08N, and Run_08, we found that the reduction in $NO_x$ emissions (17% from 2008 to 2016)) decreased the gaseous $NH_3$ concentrations by about 3% (Fig. S5). Specifically, because the reduced $NO_x$ in this period led to the transition of ozone ($O_3$) photochemistry from VOC-limited to transitional regime with high $O_3$ production efficiency (Jin and Holloway, 2015), the simulated annual mean $O_3$ concentrations were elevated by 3.7 ppb over the North China Plain between the Run_16_08N and Run_16 cases. The resultant enhancement in atmospheric oxidizing capacity would favor the conversion of $NO_2$ to $NO_3^-$ and therefore derive more $NH_3$ partitioning from gas to particle phases via aerosol thermodynamic equilibrium. Moreover, the measurements at an urban station of Beijing indicated a fluctuating trend of the annual mean $NO_3^-$ concentrations during 2013−2016 (Fig. 1). Overall, the limited reduction in $NO_x$ emissions cannot be responsible for the increased $NH_3$, because the concentrations of particulate nitrate remain high over the North China Plain during recent years."

[Figure]

**Figure 1.** (a) Inter-annual trends of $SO_2$ and $NH_3$ VCDs averaged over North China Plain from 2008 to 2016. (b) Inter-annual trends of emissions of $SO_2$ $NH_3$, and $NO_x$ in the North China Plain from 2008 to 2016, and annual mean concentrations of $PM_{2.5}$ sulfate, ammonium, and nitrate

derived from measurements at an urban station (Beijing, 39.99 °N, 116.3 ° E) in North China Plain from 2013 to 2016.

[Figure]

**Figure S5**. Absolute (a) and percent (b) changes in the simulated column concentrations of $NH_3$ between the Run_16 and Run_16_N08 ($NO_x$ emissions in 2008). Negative values denote decreases due to the change in $NO_x$ emissions in the Run_16_N08. The black box represents the major area of interest in this study.

*Similarly, for meteorological effects, quote "We also tested the effects of meteorological conditions on NH3 variations by a simulation with meteorological fields in 2016 and anthropogenic emissions in 2012 (Run_16_E12). Compared to the Run_12 case, we found the change in meteorological fields (2012 vs. 2016) had a negligible influence on NH3 concentrations in most of North China Plain." None of these were shown quantitatively! Can't make statements like these without any evidence. The following statement "Although temperature increase was reported to partly contribute to the positive trend of NH3 (Warner et 20 al., 2017; Fu et al., 2017), our simulations indicated that the overall meteorological factors could not explain the observed significant increase tropospheric NH3 concentrations over North China Plain." This sentence is misleading, as if the quoted studies were trying to explain the observed significant increase in tropospheric NH3 concentrations by meteorological factors. In fact, Warner et al. (2017) emphasized the leading cause of the NH3 increase was the reduction of SO2 in China, I quote "Over China, a combination of expanded agricultural activities, nascent SO2 control measures, and increasing temperatures cause the observed increases in ammonia."*

**Response:** Accepted. The meteorological effects were examined in this study by the simulation for 2016 with anthropogenic emissions in 2012 (there was a pronounced increase in $NH_3$ columns in the period of 2012−2016). The resulting column concentration of $NH_3$ on average over the northern China was 3% lower than that in the baseline simulation of 2012. In the area of interest, this influence on the $NH_3$ column concentrations was

minor (marked with the black box in Fig. S6). We show these quantitative results in the revised manuscript.

We agree with the referee that Warner et al. emphasized the important role of the reduction of $SO_2$ in China in the trend of $NH_3$. We cite the finding of Warner et al. (2017) to support our results.

**Revisions:** (Page 10, Line 3-11) "Besides, meteorological conditions are known to have an influence on $NH_3$ concentrations. Both Warner et al. (2017) and Fu et al. (2017) have found that elevated annual surface temperature partially contributed to the increase in $NH_3$ in East China over the past decade. In this work, we tested the effects of meteorological conditions on $NH_3$ variations by a simulation with meteorological fields in 2016 and anthropogenic emissions in 2012 (Run_12_M16). We selected these two years because $NH_3$ concentrations experienced a rapid increase during the period. This change in meteorological fields for the Run_12_M16 resulted in a decrease of about 3% in annual mean $NH_3$ concentrations relative to the Run_12 (Fig. S6). Therefore, the inter-annual variability in meteorological conditions cannot explain the observed significant increase over the North China Plain."

(Page 10, Line 12-17) "Interestingly, increasing trends of gas-phase $NH_3$ in the atmosphere have also been observed in the last twenty years in the Midwest of the United States and Western Europe by satellite retrievals and ground measurements (Warner et al., 2017; Saylor et al., 2015; Ferm and Hellstern, 2012). The marked decreases in $SO_2$ and $NO_x$ emissions were largely responsible for these increases, as confirmed by the corresponding trends of particulate sulfate and nitrate concentrations. Warner et al. (2017) infer that $SO_2$ emission reduction in China may be a leading cause of the increased $NH_3$."

[Figure]

**Figure S6**. Absolute (a) and percent (b) changes in the simulated column concentrations of $NH_3$ between the Run_12 and Run_12_16M. Negative values denote decreases due to the change in meteorological fields in the Run_12_16M. The black box represents the major area of interest in this study.

*My minor issues are mainly related to language and choice of words. I believe this manuscript needs to go through English editor at ACP. Also, many word choices are not appropriate for concise scientific publications, and somewhat wishy-washy, e.g., "appear to", "could not be", "may be a potential", "could be responsible", "would bias", ": : : concentrations disappeared", ": : : is practically zero: : :", "could result", "were almost consistent", "could make", implied", "for almost the entire: : :", "not well-regulated", "can increase: : :", "may alter": : :.*

Response: Accepted. We reworded most of these statements to make them clearer and appropriate for scientific publications. Please see the following revisions:

Revision: **Before (abbreviated as B hereafter)**: we noted that the simulated particulate nitrate ($NO_3^-$) concentrations appear to increase.
**Revision (abbreviated as R hereafter)**: (Page 9, Line 17) We noted that unlike the trend of particulate sulfate in $PM_{2.5}$, the simulated concentrations of particulate nitrate in $PM_{2.5}$ increased on average by 28% over the North China Plain between 2008 and 2016, despite a 17% reduction in $NO_x$ emissions.

B: It implied that the $NO_x$ emission reduction could not be responsible for the increase in $NH_3$.
R: (Page 10, Line 1) Overall, the limited reduction in $NO_x$ emissions cannot be responsible for the increased $NH_3$ and even had a negative contribution, because the concentrations of particulate nitrate remain high over the North China Plain during recent years.

B: although it may be a potentially important contributor to haze pollution in China.
R: (Page 2, Line 17) although they serve as an important contributor to haze pollution in China.

B: which could be responsible for such deviation between the model and observations.
R: (Page 6, Line 27) which was partially responsible for such deviation between the model and observations.

B: the relative error weighting mean method would bias a high result.
R: (Page 7, Line 11) the relative error weighting mean method always biased a high result.

B: the increasing trend of $NH_3$ column concentrations disappeared
R: (Page 7, Line 19) the increasing trend of $NH_3$ column concentrations

was not observed

B: we found that the rapid $SO_2$ emission reduction of 50% from 2012 to 2016 could result in a 55% increase in the $NH_3$ columns
R: (Page 8, Line 13) we found that the rapid $SO_2$ emission reduction of 50% from 2012 to 2016 resulted in a 55% increase in the $NH_3$ columns

B: The seasonal variations in $SO_4^{2-}$ decreases and $NH_3$ increases were almost consistent
R: (Page 8, Line 32) The seasonal variations in $SO_4^{2-}$ decreases and $NH_3$ increases were consistent

B: which could make the response of $SO_4^{2-}$ concentrations to $SO_2$ emission reductions more sensitive
R: (Page 9, Line 10) which makes the response of $SO_4^{2-}$ concentrations to $SO_2$ emission reductions more sensitive

B: It implied that the $NO_x$ emission reduction could not be responsible for the increase in $NH_3$
R: (Page 10, Line 1-2) the limited reduction in $NO_x$ emissions cannot be responsible for the increased $NH_3$ and even had a negative contribution, because the concentrations of particulate nitrate remain high over the North China Plain during recent years

B: Our work strongly indicates that the rapid $SO_2$ emission reductions (60%) from 2008 to 2016 were responsible for almost the entire $NH_3$ increases
R: (Page 10, Line 30) Our work strongly indicates that the rapid $SO_2$ emission reductions (60%) from 2008 to 2016 were responsible for the $NH_3$ increase

B: a continued increase in $NH_3$ concentrations is anticipated if $NH_3$ emissions are not well-regulated
R: (Page 11, Line 12) a continued increase in $NH_3$ concentrations is anticipated if $NH_3$ emissions are not regulated

*Page 2 line 11: "As a major agricultural country, China is the world's largest emitter of NH3: : :" what about India?*

**Response:** Accepted. The REAS2 inventory estimated the $NH_3$ emissions in India of 9.87 Tg, which is almost the same as those in China (Li et al., 2017; Kurokawa et al., 2013). We reword this sentence.

**Revisions:** (Page 2, Line 13) "As a major agricultural country, China is one of the world's largest emitters of $NH_3$."

*Page 2 line 15: ": : :may be potentially important contributor to haze: : :" It's a known fact!*

**Response:** Accepted. We rewrite this sentence.

**Revisions:** (Page 2, Line 17-18) "Until now, $NH_3$ emissions have not been regulated by the Chinese government, although they serve as an important contributor to haze pollution in China."

*Page 2 line 17-19: "Interestingly, satellite observations over the past decade have shown an increase in tropospheric columns of gaseous NH3in this area (Warner et al., 2017). But no quantitative studies have been performed to explain it." Warner et al. (2017) was a quantitative study using observations. Should be "But no sensitivity studies: : :"*

**Response:** Accepted. We reworded the sentence.

**Revisions:** (Page 2, Line 22-23) "But no sensitive studies have been performed to explain it, especially from a modelling perspective."

*Page 2 line 19-20: "Along-term bottom-up inventory indicated that NH3 emissions in China have displayed a slightly decreasing tendency." Needs references!*

**Response:** Accepted. The corresponding reference is added here.

**Revisions:** (Page 2, Line 23-25) "A long-term bottom-up inventory indicated that $NH_3$ emissions in China have displayed a slightly decreasing tendency (Kang et al., 2016)."

*Page 3 line 10: "Here, we hypothesize that the rapid SO2 emission reduction is the reason for the increase in tropospheric NH3: : :" Several studies have published the fact that the SO2 emission reduction is the reason: : :, not a hypothesis anymore. Should reference others' publications here, for global studies or in other regions, than in the North China.*

**Response:** Accepted. We provide those references in the revised manuscript.

**Revisions:** (Page 3, Line 1-3) "Several studies have proposed that reduction in $SO_2$ emissions or $NO_x$ emissions is an important factor in determining the increase in atmospheric $NH_3$ concentrations on the global and region scales (Warner et al., 2017; Yu et al., 2018; Saylor et al., 2014)."

(Page 3, Line 13-14) "Here, we hypothesize that the rapid $SO_2$ emission reduction is the main cause of the increase in tropospheric $NH_3$ concentrations over the North China Plain."

*Page 4 line 9: Please pay attention to the order when acronyms are introduced and used throughout the paper.*

**Response:** Accepted. We check the use of acronyms throughout the manuscript, including WRF-Chem, IASI, MEIC, etc.

*Page 4 line 15: MEIC should be defined on Page 3 line 7.*

**Response:** Accepted. We add a related reference for MEIC.

**Revisions:** (Page 3, Line 9) "the Multi-resolution Emission Inventory for China (MEIC) (Zheng et al., 2018)."

*Page 4 line 15: "were cut" use reduced.*

**Response:** Accepted. We reword it.

**Revisions:** (Page 4, Line 16-17) "the annual $SO_2$ emissions in North China Plain were reduced by about 60%"

*Page 4 line 19: remove "by our research group"*

**Response:** Accepted. We remove it.

**Revisions:** (Page 4, Line 20-21) "A high-resolution $NH_3$ emission inventory (1km×1km, month) was developed based on the bottom-up method."

*Page 4 line 21: "in our previous studies: : :" should be "studies by: : :"*

**Response:** Accepted. We reword the sentence.

**Revisions:** (Page 4, Line 22-23) "The full details can be found in studies by"

*Page 5 line 5-7: "Meanwhile: : :" needs references.*

**Response:** Accepted. The data about agricultural activities were shown in Table S1. The references for the source of data were shown in the supplementary file.

**Revisions:** (Page 5, Line 23-26) "On the other hand, the number of some major livestock increased (Beef −20%, Dairy +39%, Goat −23%, sheep +55%, Pig +18%, and Poultry +19%; see Table S1 for details), while the proportion of intensive animal rearing systems rises to nearly half of the livestock industry in 2016, compared to only 28% in 2008 (Table S1)."

*Page 5 line 11: use IASI.*

**Response:** Accepted. We reword the sentence.

**Revisions:** (Page 5, Line 6) "According to the measurements by IASI, the North China Plain showed the highest VCDs of $NH_3$ in China"

*Page 6 line 22: "which could be responsible", add partially responsible: :*

**Response:** Accepted. We reword the sentence in the revised paper.

**Revisions:** (Page 6, Line 26-28) "but it has not been fully included in our bottom-up inventory, which was partially responsible for such deviation between the model and observations"

*Page 6 line 23-24: bad sentence, rewrite.*

**Response:** Accepted. We rewrite it.

**Revisions:** (Page 6, Line 29-31) "We calculated the $NH_3$ VCDs from the simulations by integrating $NH_3$ molecular concentrations from the surface level to top troposphere. The results agreed well with the observed $NH_3$ columns of 2016 on the magnitude and spatial-temporal patterns (Fig. S2)."

*Page 7 line 3: "Moreover, we also: : :", remove also.*

**Response:** Accepted. We remove it.

**Revisions:** (Page 7, Line 2) "Moreover, we evaluated the modelled SNA concentrations using the filter-based $PM_{2.5}$ samples at an urban atmospheric monitoring station in North China Plain during 2014−2016."

*Page 7 line 18-19: "These tests support: : :" Too absolute! No other mechanisms?*

**Response:** Accepted. We rewrite this statement.

**Revisions:** (Page 7, Line 25-27) "Therefore, we deduce that the rapid $SO_2$ emission reductions are responsible for the increased $NH_3$ levels during 2008−2016, while other mechanisms may be negative contributors. More details on these effects are shown in the following."

*Fig. 2: use whole words for Sim., Obs., Sep., and Aug.*

**Response:** Accepted. We modify the words and the figure.

**Revisions:**

[Figure]

**Figure 2.** Comparison of modelled gaseous NH₃ concentrations with corresponding monthly measurements of NH₃ from September 2015 to August 2016. The 1:2 and 2:1 dashed lines are shown for reference and the Pearson correlation coefficient is shown inset.

*References*

Ferm, M., and Hellsten, S.: Trends in atmospheric ammonia and particulate ammonium concentrations in Sweden and its causes, Atmos. Environ., 61, 30-39, https://doi.org/10.1016/j.atmosenv.2012.07.010, 2012.

Fu, X., Wang, S., Xing, J., Zhang, X., Wang, T., and Hao, J.: Increasing Ammonia Concentrations Reduce the Effectiveness of Particle Pollution Control Achieved via SO2 and NOX Emissions Reduction in East China, Environ. Sci. Tech. let., 4, 221–227, 10.1021/acs.estlett.7b00143, 2017.

Jin, X., and Holloway, T.: Spatial and temporal variability of ozone sensitivity over China observed from the Ozone Monitoring Instrument, J. Geophys. Res. Atmos., 120, 7229-7246, doi:10.1002/2015JD023250, 2015.

Kang, Y., Liu, M., Song, Y., Huang, X., Yao, H., Cai, X., Zhang, H., Kang, L., Liu, X., Yan, X., He, H., Zhang, Q., Shao, M., and Zhu, T.: High-resolution ammonia emissions inventories in China from 1980 to 2012, Atmos. Chem. Phys., 16, 2043-2058, 10.5194/acp-16-2043-2016, 2016.

Kurokawa, J., Ohara, T., Morikawa, T., Hanayama, S., Janssens-Maenhout, G., Fukui, T., Kawashima, K., and Akimoto, H.: Emissions of air pollutants and greenhouse gases over Asian regions during 2000–2008: Regional Emission inventory in ASia (REAS) version 2, Atmos. Chem. Phys., 13, 11019-11058, 10.5194/acp-13-11019-2013, 2013.

Li, M., Zhang, Q., Kurokawa, J. I., Woo, J. H., He, K., Lu, Z., Ohara, T., Song, Y., Streets, D. G., Carmichael, G. R., Cheng, Y., Hong, C., Huo, H., Jiang, X., Kang, S., Liu, F., Su, H., and Zheng, B.: MIX: a mosaic Asian anthropogenic emission inventory under the international collaboration framework of the MICS-Asia and HTAP, Atmos. Chem. Phys., 17, 935-963, 10.5194/acp-17-935-2017, 2017.

Saylor, R., Myles, L., Sibble, D., Caldwell, J., and Xing, J.: Recent trends in gas-phase ammonia and PM2.5 ammonium in the Southeast United States, J Air Waste Manag Assoc, 65, 347-357, 10.1080/10962247.2014.992554, 2015.

Warner, J. X., Dickerson, R. R., Wei, Z., Strow, L. L., Wang, Y., and Liang, Q.: Increased atmospheric ammonia over the world's major agricultural areas detected from space, Geophys. Res. Lett., 44, 2875-2884, 10.1002/2016gl072305, 2017.

Zheng, B., Tong, D., Li, M., Liu, F., Hong, C., Geng, G., Li, H., Li, X., Peng, L., Qi, J., Yan, L., Zhang, Y., Zhao, H., Zheng, Y., He, K., and Zhang, Q.: Trends in China's anthropogenic emissions since 2010 as the consequence of clean air actions, Atmos. Chem. Phys., 18, 14095-14111, 10.5194/acp-18-14095-2018, 2018.